# Parameterizing Context: Unleashing the Power of Parameter-Efficient Fine-Tuning and In-Context Tuning for Continual Table Semantic Parsing

**Yongrui Chen**[1,2,*]   **Shenyu Zhang**[1,2,*]   **Guilin Qi**[1,2,†]   **Xinnan Guo**[1,2]

[1]School of Computer Science and Engineering, Southeast University, Nanjing 211189, China
[2]Key Laboratory of New Generation Artificial Intelligence Technology and its Interdisciplinary Applications (Southeast University), Ministry of Education, China
{yrchen, shenyuzhang, gqi}@seu.edu.cn, guoxinnan0727@163.com

## Abstract

Continual table semantic parsing aims to train a parser on a sequence of tasks, where each task requires the parser to translate natural language into SQL based on task-specific tables but only offers limited training examples. Conventional methods tend to suffer from overfitting with limited supervision, as well as catastrophic forgetting due to parameter updates. Despite recent advancements that partially alleviate these issues through semi-supervised data augmentation and retention of a few past examples, the performance is still limited by the volume of unsupervised data and stored examples. To overcome these challenges, this paper introduces a novel method integrating *parameter-efficient fine-tuning* (PEFT) and *in-context tuning* (ICT) for training a continual table semantic parser. Initially, we present a task-adaptive PEFT framework capable of fully circumventing catastrophic forgetting, which is achieved by freezing the pre-trained model backbone and fine-tuning small-scale prompts. Building on this, we propose a teacher-student framework-based solution. The teacher addresses the few-shot problem using ICT, which procures contextual information by demonstrating a few training examples. In turn, the student leverages the proposed PEFT framework to learn from the teacher's output distribution, then compresses and saves the contextual information to the prompts subsequently, eliminating the need to store any training examples. Experimental evaluations on two benchmarks affirm the superiority of our method over prevalent few-shot and continual learning baselines across various metrics.

## 1 Introduction

Tabular data is a crucial source of information for human decision-makers in various domains, including healthcare [1], retail [2] and finance [3]. Although structured query language (SQL) provides an efficient means of accessing this data, a natural language interface would make it more accessible to non-technical users. Consequently, there has been a growing interest in Table Semantic Parsing (TSP), which involves mapping natural language queries over tabular data to formal programs.

Current research [4, 5, 6] on TSP have covered many scenarios, most of which assume the datasets are static. In contrast, recent work [7] highlights the fact that new tables are often constantly emerging in response to changing circumstances, leading to a continual stream of TSP tasks. With this comes two new challenges: a) *few-shot problem* [8, 9, 10]: For a new task against unseen tables, the annotated data available in a short period is typically limited, causing the parser to be prone to over-fitting. b)

---

[*]Equal contributors.
[†]Corresponding author.

37th Conference on Neural Information Processing Systems (NeurIPS 2023).

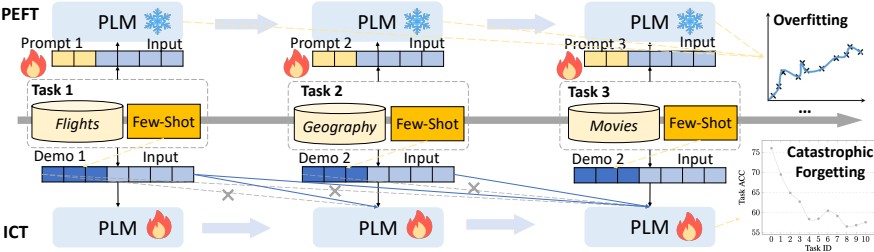

Figure 1: Processes for handling TSP task stream using PEFT (upper) and ICT (bottom). 🔥 and ❄️ represent tuning and freezing parameters, respectively. ✕ means that the previous data is not visible.

*catastrophic forgetting* [11, 7]: After training on a new task, the performance of the parser on the previous task may plummet attributed to parameter updates. While [7] mitigates these challenges using semi-supervised learning and replaying past examples, however, its performance is still limited by the amount of unsupervised data and replayed examples. More realistically, in some scenarios involving data privacy and timeliness, such as patient cases and transaction records, both unsupervised data and past examples may not be available in the following tasks.

Fortunately, recent research yields new ideas to tackle the issues. a) *in-context tuning* (ICT) [12, 13, 14] enhances models' few-shot learning capability using demonstrations containing a few training examples. b) *parameter effective fine-tuning* (PEFT) [15, 16, 17] fundamentally eliminates catastrophic forgetting by avoiding parameter updates of the *pre-trained model* (PLM) during the training stage and instead tuning a group of parameters of bearable size individually for each task. Nevertheless, the two solutions can merely handle the corresponding challenge while still suffering from the other. As is illustrated in Figure 1: for ICT, the invisibility of past demonstrations causes catastrophic forgetting when the model experiences a new task; for PEFT, though proposed to improve generalization, it still suffers from severe overfitting as only a few training examples are available for each new task.

In this paper, we propose a new method to simultaneously tackle both of these challenges by integrating PEFT with ICT. We start with a task-adaptive parameter-efficient continual TSP framework, which freezes the backbone PLM and only tunes the additional prompt embeddings for each distinct task. Unlike existing methods [16, 17], our backbone PLM undergoes fine-tuning on the initial task to facilitate adaptation to the TSP task format. Based on this foundation, we propose a *Context-Compressed Continual table semantic parser* (C3). C3 employs a teacher-student framework, comprising two parsers that symbolize the TEACHER and the STUDENT, respectively. For each task, the TEACHER utilizes ICT to glean contextual information from semantically akin demonstrations extracted from the training set, thereby mitigating the risk of overfitting on the few-shot data. Subsequently, the STUDENT adopts our proposed parameter-efficient framework to fine-tune the prompt embeddings and learns the output distribution of the TEACHER for each training example. This method ensures that the contextual information gleaned by the TEACHER can be preserved and compacted into a parameterized form, which can be reloaded later even when the demonstrations of past tasks are no longer available. Extensive experiments performed on two TSP task streams substantiate the effectiveness of our method, which surpasses all compared baselines, achieving state-of-the-art results across multiple metrics. In summary, the contributions of this paper include:

- We introduce a task-adaptive, parameter-efficient continual framework for table semantic parsing. It can completely avert catastrophic forgetting by tuning only 0.05% of the original PLM parameters.

- We propose to utilize the teacher-student framework to fuse PEFT and ICT so that the table semantic parser can benefit from both their advantages in solving few-shot data and catastrophic forgetting. To the best of our knowledge, this is the first time that the two technologies have been integrated into the scenario of the task streams.

- We create two task streams derived from two mainstream table semantic parsing benchmarks, and carry out extensive experiments using these task streams. The results demonstrate that our method surpasses existing competitors, achieving state-of-the-art performance across multiple metrics.

## 2 Preliminaries

### 2.1 Table Semantic Parsing

Formally, given a natural language question $\mathcal{Q}$ and a schema $\mathcal{S} = (\mathcal{C}, \mathcal{T})$ of multiple tables, the goal of TSP is to generate a SQL query $\mathcal{Y} = \mathcal{F}_\theta(\mathcal{Q}, \mathcal{S})$, where $\mathcal{F}_\theta$ denotes a neural semantic parser with parameters $\theta$. $\mathcal{T} = \{t_1, \ldots, t_{|\mathcal{T}|}\}$ and $\mathcal{C} = \{c_1, \ldots, c_{|\mathcal{C}|}\}$ are the sets of table names and column names, respectively, where each column $c_i \in \mathcal{C}$ belongs to only one table $t_j \in \mathcal{T}$.

In this paper, we focus on TSP in this paper for the following reasons: a) TSP is important for enhancing various fields through its role in *Natural Language Interfaces to Databases* (NLIDB). b) TSP is a challenging task requiring comprehensive understanding and mapping of queries to structured tables, presenting a few-shot learning challenge due to the difficulty in obtaining supervised data. c) TSP provides objective evaluation metrics, as minor errors in SQL queries can lead to significant outcome discrepancies, thereby offering a true reflection of our method's performance.

### 2.2 Problem Formulation

In this paper, we assume that $\mathcal{F}_\theta$ is no longer limited to fixed training and testing data, but has to face a continual stream of TSP tasks, each corresponding to a different set of tables. Formally, $\mathcal{F}_\theta$ is sequentially trained on a stream of $K$ tasks $\{\mathcal{D}^0, \mathcal{D}^1, \ldots, \mathcal{D}^{K-1}\}$. Each task $\mathcal{D}^i$ consists of a training set $\mathcal{D}^i_{\text{train}}$, a validation set $\mathcal{D}^i_{\text{valid}}$, and a test set $\mathcal{D}^i_{\text{test}}$. Each example $e = (\mathcal{X}, \mathcal{Y})$, where $\mathcal{X} = (\mathcal{Q}, \mathcal{S})$, is the input and $\mathcal{Y}$ denotes the target SQL. For $\forall \mathcal{D}^i, \forall \mathcal{D}^j$, if $i \neq j$, then $\mathcal{S}(\mathcal{D}^i) \cap \mathcal{S}(\mathcal{D}^j) = \emptyset$, where $\mathcal{S}(\mathcal{D}^i)$ denotes the total set of corresponding table schemas in $\mathcal{D}^i$. Here $\mathcal{S}$ also serves as a task identifier for each $e$, enabling efficient indexing of the corresponding task. Our ultimate objective is to train a $\mathcal{F}_\theta$ that can achieve good accuracy on each $\mathcal{D}^i_{\text{test}}$ after experiencing all $K$ tasks sequentially.

## 3 Parameter-Efficient Continual Table Semantic Parsing

We start with a vanilla parameter-efficient continual TSP framework, which trains a backbone semantic parser with a task-adaptive continual prompt-tuning method.

### 3.1 Backbone Semantic Parser

The pre-trained Text-to-Text Transfer Transformer (T5) [18] is selected as our backbone parser $\mathcal{F}_\theta$ for two primary reasons. Firstly, prior studies [16] have shown that the combination of T5 with PEFT methods can result in significant performance gains across various NLP tasks. Secondly, T5 has demonstrated robust capabilities [19] in generating logical forms on a widely-used TSP dataset Spider [5]. In particular, each input pair $\mathcal{X} = (\mathcal{Q}, \mathcal{S})$ is flattened into a plain text,

$$\mathcal{X}' = t_1 : c_1^{t_1}, c_2^{t_1}, \ldots, c_{m_1}^{t_1}; t_2 : c_1^{t_2}, c_2^{t_2}, \ldots, c_{m_2}^{t_2}; \ldots | \mathcal{Q}, \tag{1}$$

where $c_j^{t_i}$ denotes the $j$-th column name of the $i$-th table, and ":", ",", and "|" are predefined separators. Subsequently, $\mathcal{X}'$ is fed to the parser $\mathcal{F}_\theta$ and the generative probability of a predicted SQL $\tilde{\mathcal{Y}}$ is estimated by $P(\tilde{\mathcal{Y}}|\mathcal{X}', \theta) = \prod_{j=1}^{|\tilde{\mathcal{Y}}|} P(\tilde{y}_j|\mathbf{X}', \tilde{y}_{<j}, \theta)$, where $y_j \in \mathcal{V}$ denotes the $j$-th token of $\tilde{\mathcal{Y}}$ and $\mathcal{V}$ is the vocabulary of the T5 tokenizer. $\mathbf{X}'$ are the hidden states of $\mathcal{X}'$ obtained by the T5 encoder, and $P(\tilde{y}_j|\mathbf{X}', \tilde{y}_{<j}, \theta)$ denotes the output probability of $\tilde{y}_j$ from the T5 decoder.

### 3.2 Task Adaptive Continual Prompt Tuning

Prompt tuning [20], as a representative of PEFT-based methods, has been widely applied in recent NLP tasks [16, 17] owing to its simplicity. Inspired by these works, we employ prompt tuning to conduct the PEFT training framework for $\mathcal{F}_\theta$. Formally, each input $\bar{\mathcal{X}}$ in task $\mathcal{D}^k$ is prefixed with $M$ special prompt tokens $\mathcal{P}^k = p_1^k p_2^k \ldots p_M^k$. During training, only the embeddings of these tokens are updated, rather than the parameters of the entire T5 model. More specifically, the training process can be divided into the following two phases.

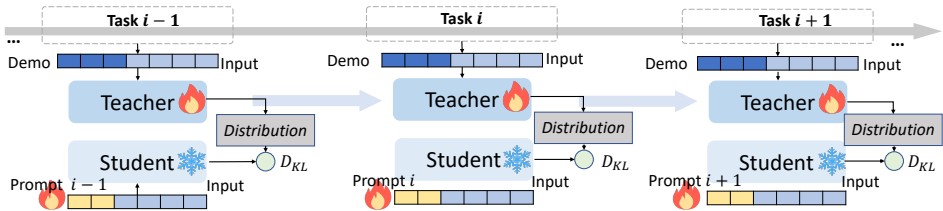

Figure 2: Illustration of the proposed C3 table semantic parser.

### 3.2.1 Initial Task Adaptation

Intuitively, the original T5 is not well-suited to TSP, as it is not exposed to any logical form during pre-training and is never tasked with predicting structured outputs. While this limitation can be readily addressed through fine-tuning, we hypothesize that tuning prompts alone may not be sufficient.

Therefore, we first fine-tune T5 with $\mathcal{D}^0_{\text{train}}$ to obtain a set of task-adaptive parameters $\theta$ that will be frozen on $\mathcal{D}^1, \ldots, \mathcal{D}^{K-1}$. Notably, here the prompt embeddings are tuned as well as $\theta$ because we expect to acquire a good initialization for both of them. In particular, for each $e = (\mathcal{X}, \mathcal{Y}) \in \mathcal{D}^0_{\text{train}}$, the following loss is optimized using the AdaFactor [21] algorithm,

$$\mathcal{L}(e; \theta, \mathbf{P}^0) = -\log P(\mathcal{Y}|\mathcal{X}', \theta) = -\sum_{j=1}^{|\mathcal{Y}|} \log P(y_j | [\bar{\mathbf{P}}^0; \mathbf{X}'], y_{<j}, \theta), \tag{2}$$

where $[;]$ represents the concatenation of matrices, $\mathbf{P}^0 \in \mathbb{R}^{M \times d}$ denotes the tunable embeddings of $\mathcal{P}^0$, and $\bar{\mathbf{P}}^0 \in \mathbb{R}^{M \times d}$ represents its hidden states from the T5 encoder. In our experiments, $\mathbf{P}^0$ is randomly initialized. After multiple iterations, the pair of $\theta$ and $\mathbf{P}^0$ with the highest validation accuracy, denoted by $(\theta^*, \mathbf{P}^*)$, is selected for initializing the subsequent $K - 1$ tasks.

### 3.2.2 Continual Prompt Tuning

Unlike existing methods [22], for $\mathcal{D}^i$ ($0 < i < K$), we initialize all $\mathbf{P}^i$ with $\mathbf{P}^*$ instead of $\mathbf{P}^{i-1}$ because our experimental results (detail in Section 5.3.2) reveal that the latter does not yield significant performance improvements. Furthermore, this strategy enhances the flexibility of the proposed framework by eliminating the dependency on the order of subsequent tasks, as all prompts share a common initialization. During training, $\mathbf{P}^i$ is updated with the gradient $\nabla_{\mathbf{P}^i} \sum_{e \in \mathcal{D}^i_{\text{train}}} \mathcal{L}(e; \theta^*, \mathbf{P}^i)$ using the AdaFactor optimizer, where $\mathcal{L}(e; \theta^*, \mathbf{P}^i)$ is calculated by Equ. (2). We experimentally set the prompt dimension $d = 512$ and the prompt number $M = 150$, leading to a total of only 0.7 million parameters. This size is negligible (0.05%) compared to the entire T5-BASE of 220 million parameters. Consequently, the cost of storing all $\mathbf{P}^1, \ldots, \mathbf{P}^{K-1}$ is usually acceptable. During inference, given a test example $e^i_{\text{test}} \in \mathcal{D}^{0:K-1}_{\text{test}}$, its task index $i$ can be first identified using its table schema $\mathcal{S}$. Then, $\mathcal{F}$ can solve it by loading the saved $(\mathbf{P}^i, \theta^*)$ without any forgetting.

In theory, our framework can also be expanded to the tasks whose tables have not been seen in $\mathcal{D}^0, \ldots, \mathcal{D}^{K-1}$. The prediction can be made using simple heuristics, such as averaging all $\mathbf{P}^i$. Since this scenario involves the zero-shot learning problem, we leave it for future work.

## 4 C3 Semantic Parser

Figure 2 depicts the illustration of our proposed *Context-Compressed Continual table semantic parser* (C3). C3 is typically composed of two parsers (e.g., T5) of identical architecture (optional), namely TEACHER $\mathcal{F}_{\text{tea}}$ and STUDENT $\mathcal{F}_{\text{stu}}$, and each possesses its own individual parameters during training.

### 4.1 Context-Enhanced Teacher Parser

The sole purpose of $\mathcal{F}_{\text{tea}}$ is to gain the few-shot learning capability and thereby *in-context tuning* (ICT) is employed to train $\mathcal{F}_{\text{tea}}$. Note that $\mathcal{F}_{\text{tea}}$ only focuses on each current task $\mathcal{D}^i$ and does not care if the past tasks are forgotten. Thus, it is sequentially trained on $\mathcal{D}^0, \ldots, \mathcal{D}^{K-1}$ without any continual

learning. During $\mathcal{D}^i$, the parameters of $\mathcal{F}_{\text{tea}}$, denoted by $\theta_{\text{tea}}^i$, is initialized with $\theta_{\text{tea}}^{i-1}$. Afterwards, $\theta_{\text{tea}}^i$ is updated by optimizing $\mathcal{L}(x; \theta_{\text{tea}}) = -\sum_{j=1}^{|\tilde{\mathcal{Y}}|} \log P(\tilde{y}_j | \mathbf{C}, y_{<j}, \theta_{\text{tea}})$. Here $\mathbf{C}$ denotes the hidden states of the input text, $\mathcal{C}$, which is constructed by retrieving and formatting the demonstrations.

### 4.1.1 Demonstration Retrieval

The key to ICT lies in identifying the suitable demonstration that effectively facilitates the learning process of the model within the appropriate context. Recent studies [23, 24] have reached a consensus that the closer the demonstration aligns with the example being queried, the greater the benefits of ICT. Thus, for each $e$, we retrieve the demonstration by comparing its semantic similarity to the candidate demonstrations. To maintain overall uniformity, we select the original T5 checkpoint which was fine-tuned on the Semantic Textual Similarity Benchmark (STS-B) [25], as an off-the-shelf retriever $\mathcal{R}$. Specifically, for example $e$ to be queried and a candidate demonstration $e_j' \in \mathcal{D}_{\text{train}}^i \setminus \{e\}$, we first wrap them into a template $T(e, e_j') = \texttt{stsb sentence1:}\mathcal{Q}, \texttt{ sentence2:}\mathcal{Q}_j'$, where $\mathcal{Q}$ and $\mathcal{Q}_j'$ represent the natural language questions of $e$ and $e_j'$, respectively. Here the target SQL $\mathcal{Y}$ is not considered because it is invisible in the testing stage. Subsequently, the similarity score of $e_j'$, denoted by $\sigma(e_j')$, is calculated by $\sigma(e_j') = \mathcal{R}(T(e, e_j')) \in [0, 5]$. To circumvent the noise, we set a threshold $\eta$ and eventually select the top-$r$ $e_j' \in \mathcal{D}_{\text{train}}^i \setminus \{e\}$ whose $\sigma(e_j') \geq \eta$ as the demonstration of $e$. In fact, not every example $e \in \mathcal{D}_{\text{train}}^i$ has sufficient demonstrations. To retain $\mathcal{F}_{\text{tea}}^i$'s ability in handling no-demonstration examples, we apply a simple but effective *context mixing* strategy during training: both $\mathcal{C}$ and $\mathcal{X}'$ are engaged in the training for each $e \in \mathcal{D}_{\text{train}}^i$ if it has $r$ demonstrations, otherwise only $\mathcal{X}'$ is engaged.

### 4.1.2 Demonstration Formatting

$r$ retrieved demonstrations $e_1, \ldots, e_r$ compose the input $\mathcal{C} = \mathcal{Q}_1|\mathcal{Y}_1|\ldots|\mathcal{Q}_r|\mathcal{Y}_r||\mathcal{X}'$, where both $|$ and $||$ are separators, and $\mathcal{X}'$ is obtained by Equ. (1). The decision to exclude table schemas $\mathcal{S}$ from $\mathcal{C}$ is motivated by one main consideration: the selected demonstration examples and the example $e$ to be queried are likely to be associated with the same batch of tables due to their similarity. Therefore, the explicit information regarding schemas need not be provided repeatedly outside of $\mathcal{X}'$, which helps to streamline the input to improve efficiency.

## 4.2 Context-Compressed Student Parser

$\mathcal{F}_{\text{stu}}^i$ follows the PEFT framework introduced in Section 3 and is dedicated to compressing and preserving the few-shot capability $\mathcal{F}_{\text{tea}}^i$ learns from the demonstration into the prompt embeddings. In this way, $\mathcal{F}_{\text{stu}}^i$ can reproduce the few-shot performance by loading the saved prompt embedding, without accessing any examples or demonstrations of previous tasks. Inspired by [26], We hypothesize that this goal could be accomplished as much as possible by having $\mathcal{F}_{\text{stu}}^i$ learn the $\mathcal{F}_{\text{tea}}^i$'s output distribution. More concretely, we minimize the KL divergence between $\mathcal{F}_{\text{stu}}^i$'s per-step autoregressive distribution and the $\mathcal{F}_{\text{tea}}^i$'s, denoted by $D_{\text{KL}}(P(\theta_{\text{tea}}) \parallel P(\theta_{\text{stu}}))$ where $P(\theta) = P(y_j | \mathbf{C}, y_{<j}, \theta)$. During training, the *teacher-forcing* strategy is applied to ensure that the inputs to the decoders of $\mathcal{F}_{\text{tea}}^i$ and $\mathcal{F}_{\text{stu}}^i$ are consistent at each decoding step. In addition, to guarantee the quality of supervision, we directly use the gold SQL $\mathcal{Y}$ instead of $P(\theta_{\text{tea}})$ if $\mathcal{F}_{\text{tea}}^i$'s prediction for a given input $\mathcal{X}$ is incorrect.

# 5 Experiments

## 5.1 Experimental Setup

### 5.1.1 Datasets & Evaluation Metrics

Our experiments involve two widely-used TSP datasets: **WikiSQL** [4] is a dataset for single-table semantic parsing in which each SQL query corresponds to a single table and adheres to a simplified SQL syntax. **Spider** [5] is a more challenged dataset in which each SQL requires the `JOIN` operation of multiple tables and contains complicated syntax including `GROUP BY` even nested query.

As explicated in Section 2.2, we constructed two task streams based on the above datasets. The first one, **Spider-Stream**, is established by merging the Spider training and validation sets and splitting

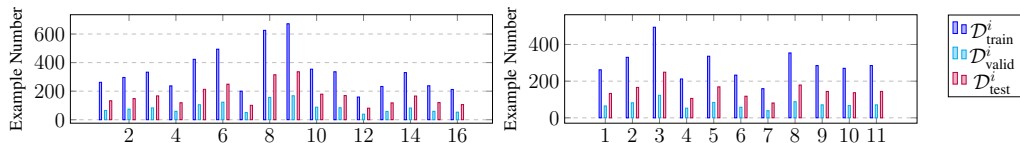

Figure 3: Statistics of the task splits of Spider-Stream (left) and Combined-Stream (right).

them into 16 tasks according to the domain of the tables. It is used to reflect the impact of the changing table domains. The second one, **Combined-Stream**, is a merged task stream encompassing 11 tasks. The first 8 tasks are randomly selected from the Spider-Stream and encompass complex SQL syntax, whereas the last 3 tasks are collected from WikiSQL and involve only simple SQL syntax. It is utilized to assess the dual impact of table domains and target SQL structures.

To highlight the few-shot challenge, the majority of tasks $\mathcal{D}_{\text{train}}^i$ have $|\mathcal{D}_{\text{train}}^i| < 500$. For each method, we evaluated its performance in three random task orders, to eliminate any potential influence of specific tasks. Considering that in real applications, models are usually fully trained in certain domains before facing the task stream, we combined the first $N$ (6 for Spider-Stream and 5 for Combined-Stream) tasks of each task order into a single task. This practice enables all methods to have full adaptation to the TSP task format. In addition, for Combind-Stream, to aggravate the gap between adjacent tasks, for any task order, we always make the tasks of the even index (0, 2, ...) from Spider and the tasks of the odd index (1, 3, ...) from WikiSQL.

Following [27, 11], we adopted four metrics: 1) Task Accuracy: TA $= \frac{1}{K} \sum_{i=0}^{K-1} a_{i,K-1}$; 2) Example Accuracy: EA $= a_{\mathcal{D}_{\text{test}}^{(0:K-1)}}$; 3) Initial Accuracy: IA $= \frac{1}{K} \sum_{i=0}^{K-1} a_{i,i}$; 4) Memory Decay: MD $= \frac{1}{K-1} \sum_{i=0}^{K-2} a_{i,K-1} - a_{i,i}$; where $a_{i,j}$ denotes the accuracy on $\mathcal{D}_{\text{test}}^i$ after the training of $\mathcal{D}^j$. TA and EA reflect the overall performance at the task level and example level, respectively. MD measures the extent of forgetting. IA evaluates the performance when the parser is trained on a new task for the first time, considering no forgetting.

### 5.1.2 Compared Methods & Implementation Details

Our compared methods can be categorized into three groups: a) FINE-TUNING, which trains the parser sequentially without any strategies; b) **Few-Shot Learning** baselines, which contain MAML [8] and ICT [13]; c) **Continual Learning** baselines, which consist of EWC [28], HAT [29], EMR [27], EMAR [30], APPER [31], and TR [11]. To increase the competitiveness of these baselines, we utilized pre-trained GRAPPA-LARGE [32] for TSP in conjunction with SemQL [33], an intermediate representation for SQL. To ensure a fairer comparison, we conducted experiments applying the best-performing methods on GRAPPA, EMR and EMAR, to T5-BASE as complementary baselines. Additionally, we compared our methods with SFNet [7] which utilizes extra unsupervised data. Note that we only apply [7] on Combined-Stream for the reason that Combined-Stream randomly selects a portion of samples from the original Spider and WikiSQL datasets which enables the rest to be used as unsupervised data to fulfill the scenario requirements. We also set an upper boundary method MULTI-TASK. For each task $\mathcal{D}^i$, it trains $\mathcal{F}_\theta$ with $\mathcal{D}_{\text{train}}^{(0:i)}$.

Our method ran on one NVIDIA RTX 4090 GPU and the hyperparameters were decided with the validation sets (See Appendix for details): a) prompt length $M$ is set to 150; b) demonstration number $r$ and threshold $\eta$ are set to 1 and 4.0, respectively; c) the batch size is 12 and the learning rates are set to 0.3 and $1 \times 10^{-4}$ for prompt-tuning and fine-tuning, respectively; d) early stopping is performed after 1000 epochs and the patience is set to 10 evaluations, which is performed every 50 epochs. e) T5-LARGE is always employed as $\mathcal{F}_{\text{tea}}$. All of our data and codes are publicly available[1].

### 5.2 Overall Results

Table 1 presents the overall performance of the methods. The results indicate that most of the compared methods exhibit relatively lower effectiveness and more severe forgetting on Combined-Stream compared to Spider-Stream, even though their backbone models have been pre-trained for the

---

[1]`https://github.com/Bahuia/C3`

Table 1: Experimental results for comparison with baselines in 3 random task orders. Means and standard variations are reported. The absence of standard deviation for PEFT and C3 is due to the fact that their performance is order-independent. ♠ indicates using the replayed memory of size 15 and ♣ indicates using additional unsupervised data.

| Backbone | Method | Spider-Stream | | | Combined-Stream | | |
|---|---|---|---|---|---|---|---|
| | | TA (%) | EA (%) | MD (%) | TA (%) | EA (%) | MD (%) |
| GRAPPA-LARGE (340M) | FINE-TUNING | $56.9_{1.0}$ | $54.6_{1.0}$ | $-18.8_{1.5}$ | $37.6_{1.8}$ | $43.9_{0.9}$ | $-39.1_{2.2}$ |
| | MAML [8] | $52.2_{1.3}$ | $49.1_{1.5}$ | $-19.5_{2.2}$ | $31.3_{1.3}$ | $37.2_{1.4}$ | $-43.8_{1.5}$ |
| | ICT [13] | $57.0_{1.4}$ | $54.3_{2.2}$ | $-17.1_{2.1}$ | $37.9_{4.0}$ | $43.9_{1.8}$ | $-37.4_{4.4}$ |
| | EWC [28] | $57.5_{3.3}$ | $55.1_{2.4}$ | $-17.7_{3.9}$ | $37.0_{1.9}$ | $44.1_{0.9}$ | $-38.4_{2.2}$ |
| | HAT [29] | $57.8_{2.9}$ | $54.8_{3.4}$ | $-17.0_{3.3}$ | $38.5_{4.1}$ | $45.0_{2.0}$ | $-37.6_{5.5}$ |
| | EMR♠ [27] | $65.2_{0.2}$ | $62.9_{0.6}$ | $-9.4_{0.8}$ | $60.9_{0.7}$ | $58.6_{1.8}$ | $-10.3_{1.8}$ |
| | EMAR♠ [30] | $62.8_{1.2}$ | $60.8_{1.2}$ | $-10.5_{1.1}$ | $63.1_{1.8}$ | $60.8_{0.9}$ | $-7.7_{2.6}$ |
| | APPER [31] | $57.9_{1.6}$ | $55.8_{1.7}$ | $-17.2_{2.6}$ | $37.1_{2.4}$ | $44.0_{0.6}$ | $-38.3_{2.9}$ |
| | TR♠ [11] | $57.9_{1.2}$ | $55.1_{1.4}$ | $-15.8_{1.5}$ | $59.7_{1.0}$ | $56.3_{1.1}$ | $-11.9_{0.7}$ |
| | SFNET♣ [7] | - | - | - | $60.7_{0.9}$ | $57.0_{1.9}$ | $-6.0_{1.2}$ |
| T5-BASE (220M) | EMR♠ [27] | $60.3_{0.9}$ | $56.6_{0.1}$ | $-13.4_{0.2}$ | $62.6_{0.4}$ | $60.0_{1.4}$ | $-6.5_{0.6}$ |
| | EMAR♠ [30] | $57.2_{0.4}$ | $52.7_{1.0}$ | $-16.7_{0.8}$ | $62.1_{0.2}$ | $58.1_{1.1}$ | $-6.5_{0.3}$ |
| | PEFT | $65.7$ | $64.5$ | $0.0$ | $63.8$ | $66.2$ | $0.0$ |
| | C3 | $67.5_{0.3}$ | $66.5_{0.2}$ | $0.0_{0.0}$ | $66.3_{0.2}$ | $67.6_{0.2}$ | $0.0_{0.0}$ |
| | MULTI-TASK | $76.3_{0.5}$ | $76.2_{1.0}$ | $3.2_{0.7}$ | $70.0_{0.9}$ | $71.1_{0.7}$ | $1.7_{0.2}$ |
| T5-LARGE (770M) | PEFT | $69.8$ | $67.4$ | $0.0$ | $67.3$ | $70.0$ | $0.0$ |
| | C3 | $\mathbf{71.1_{0.3}}$ | $\mathbf{69.7_{0.5}}$ | $\mathbf{0.0_{0.0}}$ | $\mathbf{68.3_{0.5}}$ | $\mathbf{70.6_{0.4}}$ | $\mathbf{0.0_{0.0}}$ |

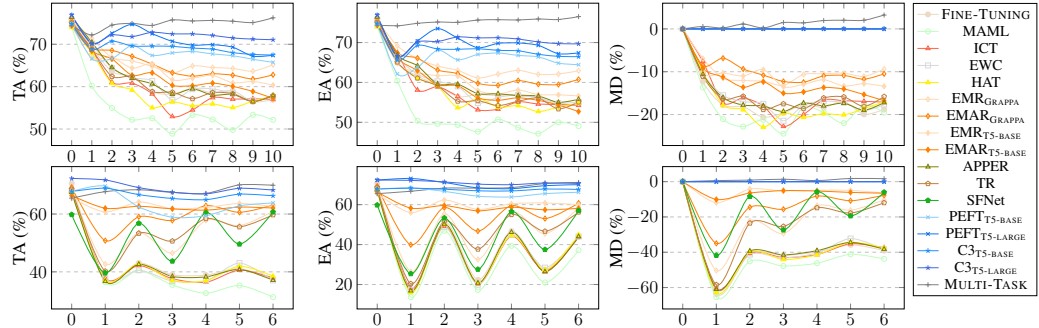

Figure 4: TA (%), EA (%), and MD (%) till the seen tasks of Spider-Stream (upper) and Combined-Stream (bottom) after learning on each task. Only the means are reported in 3 random task orders.

tabular data. Surprisingly, our proposed PEFT framework outperforms all baseline models despite the absence of any explicit provision to address the few-shot problem, mainly due to its ability to retain previously learned knowledge. Moreover, our proposed C3 further improves the overall performance (4.9% & 5.1% in terms of TA) and achieves state-of-the-art results for both datasets. Of particular note is that even with a smaller backbone model (T5-BASE, 220M), C3 outperforms all compared methods based on GRAPPA-LARGE (340M). Remarkably, it manages to approach the upper bound of performance (MULTI-TASK) on Combined-Stream.

Compared to ICT, MAML performs poorly on both datasets, especially in terms of MD, even less than FINE-TUNING. This finding may point out that it is more profound and irreversible for parameter updates. The replay-based EMR and EMAR outperform other continual learning baselines in terms of all metrics, which suggests that retaining past instances may be more direct and effective for mitigating forgetting, particularly on Combined-Stream where the task varies more. Unfortunately, in many cases involving data privacy, access to past examples is disabled, rendering these methods limited. Conversely, our proposed C3 not only circumvents data privacy concerns and has a broader range of applications, but also achieves superior results without using any past examples.

Table 2: Experimental results of ablation studies in the same task order. ♣ indicates that the backbone used is T5-LARGE, since only TEACHER remains when STUDENT is removed.

| Backbone | Method | Spider-Stream | | | Combined-Stream | | |
|---|---|---|---|---|---|---|---|
| | | TA (%) | EA (%) | MD (%) | TA (%) | EA (%) | MD (%) |
| T5-BASE | FINE-TUNING | 53.3 | 50.8 | −20.7 | 59.1 | 58.2 | −9.1 |
| | C3 | 67.7 | 66.7 | 0.0 | 66.4 | 67.7 | 0.0 |
| | w/o TEACHER | 65.7 | 64.5 | 0.0 | 63.8 | 66.2 | 0.0 |
| | w/o STUDENT ♣ | 64.9 | 62.8 | −14.6 | 59.6 | 59.6 | −14.9 |
| | w/o In-Context Tuning | 65.6 | 64.3 | 0.0 | 65.3 | 66.9 | 0.0 |
| | w/o Task Adaptation | 19.4 | 20.6 | 0.0 | 35.8 | 25.7 | 0.0 |
| | w Continual Initialization | 67.0 | 65.4 | 0.0 | 66.0 | 67.4 | 0.0 |
| T5-LARGE | FINE-TUNING | 60.5 | 56.3 | −17.3 | 65.3 | 67.2 | −8.8 |
| | C3 | **70.7** | **68.9** | **0.0** | **69.0** | **71.2** | **0.0** |

Table 3: Performance of C3 using GPT as the TEACHER parser.

| STUDENT | TEACHER | Spider-Stream | | Combined-Stream | |
|---|---|---|---|---|---|
| | | TA (%) | EA (%) | TA (%) | EA (%) |
| T5-BASE | text-davinci-003 | 66.3 | 64.8 | 65.5 | 66.8 |
| | T5-LARGE | **67.5**$_{0.3}$ | **66.5**$_{0.2}$ | **66.3**$_{0.2}$ | **67.6**$_{0.2}$ |
| T5-LARGE | text-davinci-003 | **71.3** | **69.6** | 67.6 | 70.0 |
| | T5-LARGE | 71.1$_{0.3}$ | 69.7$_{0.5}$ | **68.3**$_{0.5}$ | **70.6**$_{0.4}$ |

## 5.3 Detailed Results and Analysis

### 5.3.1 Performance Till the Seen Tasks

Figure 4 displays the methods' performance till the seen tasks on both datasets. Our proposed C3-LARGE (blue) consistently exhibits the highest performance across all evaluated time steps. Moreover, this performance advantage becomes increasingly pronounced as the number of tasks in the continual learning regime grows. Interestingly, the EA metrics on the Combined-Stream for compared methods always show oscillation. The performance troughs on the even-indexed tasks suggest a dramatic forgetting of the harder Spider tasks after the model is trained on an easier WikiSQL task.

### 5.3.2 Ablation Study

We compared the performance of the proposed C3 with the following settings:

- **w/o TEACHER**: we removed $\mathcal{F}_{\text{tea}}$ to verify its contribution;
- **w/o STUDENT**: we removed $\mathcal{F}_{\text{stu}}$ and only employed $\mathcal{F}_{\text{tea}}$ (T5-LARGE) to perform ICT, to evaluate the contribution of PROMPT-TUNING;
- **w/o In-Context Tuning**: we used FINE-TUNING to train $\mathcal{F}_{\text{tea}}$ to assess the necessity of ICT.
- **w/o Task Adaptation**: we used PROMPT-TUNING to train $\mathcal{F}_{\text{stu}}$ from $\mathcal{D}^0$.
- **w Continual Initialization**: we initialize the prompt embeddings $\mathbf{P}^i$ with $\mathbf{P}^{i-1}$ instead of $\mathbf{P}^*$.

Table 2 shows the TA (%), EA (%), and MD (%) of different settings. Our C3 equipped with all components performs best in terms of all metrics. Dramatic performance degradation demonstrates that the adaptation of the PLM to the task format is critical to the effectiveness of prompt tuning. Removing $\mathcal{F}_{\text{tea}}$ results in an approximate 2% drop in terms of both TA and EA, which proves the TEACHER's few-shot learning capability contributes to the entire model. With the exclusion of STUDENT, the TA exhibits a more pronounced decrease on Combined-Stream (-6.8%) compared to Spider-Stream (-2.8%) because Combined-Stream involves challenges of both domain changes and SQL structure changes. The performance drops (-2.3% & -0.7%) brought by abandoning ICT reveal the necessity of contextual information. Its smaller contribution on Combined-Stream is probably due to the relative simplicity of the WikiSQL tasks, which does not require a demonstration to make

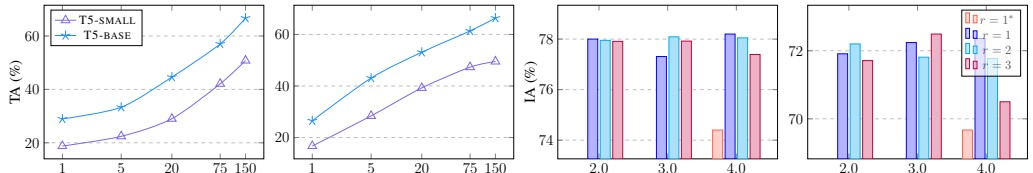

Figure 5: Left two: performance of C3 with different prompt lengths $M$ and PLM sizes; Right two: IA (%) of the TEACHER parser with different thresholds $\eta$ and demonstration numbers $r$, where $*$ indicates the absence of the context mixing strategy (detailed in Section 4.1.1).

a correct prediction. In contrast to [16]'s assertion, our experiments did not show any improvement with Continual Initialization. One possible reason is the large gap between different tasks.

### 5.3.3 Impact of Large Language Models

Considering the amazing potential of Large Language Models (LLMs), such as GPT-3 [34] and GPT-4 [35], in the NLP community recently, we also evaluate the performance of C3 by replacing $\mathcal{F}_{\text{tea}}$ from T5-LARGE with the text-davinci-003, an LLM of the GPT family that excels in textual tasks. Here we did not choose code-davinci-002, which specifically handles code tasks, and the more widely used gpt-3.5-turbo, because the API of the former is deprecated and the latter can not provide output distribution for each decoding step. Referring to [26], we align the GPT tokenizer with the T5 tokenizer in order to calculate $D_{\text{KL}}$ (details in Appendix). Here we let text-davinci-003 perform inference based on the extracted demonstration without tuning its parameters, so C3 in this scenario is independent of the task order. The experimental results are shown in Table 3. Despite not having undergone any fine-tuning, using GPT has achieved close or even better results compared to using T5-LARGE. This proves, to some extent, that our proposed method is not limited to a specific architecture (encoder-decoder vs. decoder-only) and size (million-level vs. billion-level) of PLM.

### 5.3.4 Impact of Different Prompt Lengths & PLM Sizes

We varied the backbone PLM in {T5-SMALL, T5-BASE} and prompt length $M$ in {1, 5, 20, 75, 150} for our proposed C3 on the two task streams. The experimental results are shown in Figure 5, which indicates that: When fixing the prompt length, increasing the backbone PLM size improves the TA; When the PLM size is fixed, increasing the prompt length improves the overall performance in general. Moreover, our findings indicate that extending the prompt length from 20 to 75 enhances TA more significantly than increasing it from 75 to 150 on both task streams.

### 5.3.5 Impact of Different Demonstration Numbers & Thresholds

We also varied the demonstration number $r$ in {1, 2, 3} and threshold $\eta$ in {2.0, 3.0, 4.0} for the TEACHER $\mathcal{F}_{\text{tea}}$ in our proposed C3. Figure 5 shows IA (%). We observed that removing the context mixing strategy leads to severe performance degradation, proving our hypothesis that there exists many examples lack of sufficient demonstrations. When using context mixing, the performance is not significantly affected by the threshold and number of demonstrations, but a trade-off between performance and efficiency is achieved with a threshold of 4.0 and using only one demonstration.

### 5.3.6 Performance of TEACHER Parser with Always-Visible Demonstrations

While in many works involving in-context learning there exists an always-visible demonstration pool (e.g., public or synthetic datasets) from which we can select demonstration for all tasks, resources for TSP tasks are quite limited due to high expense of annotation. Therefore, C3 applies a practical strategy which samples demonstrations from every incoming task. However, to study the above sampling strategy, we conducted an additional experiment assuming that $\mathcal{D}_{\text{train}}^0$, the training set for task 0, is an always-visible public demonstration pool from which all demonstrations for subsequent tasks are sampled. Table 4 shows IA (%) of a T5-LARGE based teacher parser. From the results, when sampling the demonstration from $\mathcal{D}_{\text{train}}^0$, the performance of the teacher parser shows a significant drop on both datasets. This reflects the conclusions in [23, 24], which state that performance gains are maximized only when in-context learning utilizes demonstrations similar to the test examples.

Table 4: IA (%) of TEACHER Parser with Always-Visible Demonstrations.

| TEACHER | Spider-Stream | Combined-Stream |
|---|---|---|
| Demonstrations from $\mathcal{D}_{\text{train}}^0$ | 76.3 | 70.6 |
| C3 | **78.2** | **72.4** |

For domain-specific databases, there is no guarantee that similar demonstrations will be found using public data resources. Our results prove that that the quality of the demonstration has a great impact on the performance of ICT.

## 6 Related Work

**Table Semantic Parsing** In recent years, there has been a notable increase in interest concerning joint textual-tabular data understanding problems [36, 37], especially in table semantic parsing (TSP) [4, 5]. This trend can be attributed to the fact that many real-world datasets contain a combination of structured and unstructured data, and a comprehensive understanding of such data requires the ability to analyze and interpret both forms effectively. The related methods can be divided into three directions according to the application scenarios, namely single table [4, 38, 10, 39], multi-table [5, 33, 40, 41] and conversational tasks [42, 6, 43, 22].

**Parameter Efficient Continual Learning** Prompt tuning was first systematically studied by [20], who demonstrated that fine-tuning language models on a small number of task-specific input prompts could result in substantial performance improvements with much lower computational costs compared to full-model fine-tuning. Afterward, several works [44, 17, 16, 45] applied it to the continual learning scenario and achieved promising results for mitigating catastrophic forgetting. Unlike them, we leverage in-context tuning to compensate for the shortcomings of prompt tuning on few-shot tasks.

**Few-shot In-Context Learning** Recently, in-context learning has emerged as a promising approach to improve the performance of large language models (LLMs) such as GPT-3, particularly in scenarios where only limited training data is available [34]. However, LLMs are often computationally expensive to train and inference, which limits their practical use in many real-world applications. To address this limitation, recent studies [12, 13, 14] focus on extending the capabilities of in-context learning to medium- or small-scale pre-trained language models (PLMs) by tuning them with the demonstrations. Inspired by these works, we propose to utilize ICT in a parameter-efficient framework as a means to acquire contextual information.

## 7 Conclusion & Future Work

In this paper, we present a few-shot continual learning method that integrates *parameter-efficient fine-tuning* (PEFT) and *in-context tuning* (ICT) to handle the task stream in table semantic parsing. The teacher model is designed to extract contextual information from sampled demonstrations for few-shot learning, while the student model is tasked with learning the teacher's output distribution. It employs parameter-efficient prompt-tuning to retain its capabilities and to entirely eradicate catastrophic forgetting. Through extensive experimentation, we've demonstrated that our method surpasses all compared baselines. In our future research, we aim to investigate strategies to ensure the model's zero-shot predictive capabilities, while simultaneously preventing catastrophic forgetting. Theoretically, our proposed method is generalizable and has the potential to be adapted to other NLP tasks because it is not limited to specific backbone models and does not involve task-specific modules. We leave the evaluation of C3 on other tasks to our future work.

**Acknowledgements and Disclosure of Funding**

This work is partially supported by the Natural Science Foundation of China (Grant No. U21A20488). We thank the Big Data Computing Center of Southeast University for providing the facility support on the numerical calculation in this paper.

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

# A Hyperparameter Details

The detailed settings of the hyper-parameters used in our experiments are presented in Table 5.

Table 5: Hyper-parameters settings.

| Method | Symbol | Value | Description |
|--------|--------|-------|-------------|
| C3 | $M$ | 150 | # of prompt tokens |
| | $r$ | 1 | # of demonstrations |
| | $\eta$ | 4.0 | threshold |
| | - | 12 | batch size |
| | - | 0.3 | learning rate for PROMPT-TUNING |
| | - | 4 | beam size |
| | - | $1 \times 10^{-4}$ | learning rate for FINE-TUNING |
| | - | 50 | evaluation is done every # epochs for PROMPT-TUNING of T5-BASE or T5-SMALL |
| | - | 25 | evaluation is done every # epochs for PROMPT-TUNING of T5-LARGE |
| | - | 10 | evaluation is done every # epochs for FINE-TUNING of T5-BASE or T5-SMALL |
| | - | 5 | evaluation is done every # epochs for FINE-TUNING of T5-LARGE |
| | - | 15000 | # of maximum epochs for PROMPT-TUNING |
| | - | 300 | # of maximum epochs for FINE-TUNING |
| | - | 512 | maximum # of input tokens |
| | - | 10 | stop training when the specified metric worsens for # evaluations |
| Other Baselines | - | 32 | batch size |
| | - | $2 \times 10^{-4}$ | learning rate |
| | - | 15 | memory size of replayed examples in EMR, EMAR, TR and EWC |
| | - | 5 | beam size |
| | - | 55 | # of maximum epochs |
| | - | 1 | # of maximum epochs for the second iterations in EMAR |
| | - | 2 | N-way of MAML |
| | - | 1 | K-shot of MAML |
| | - | 25 | # of meta-task in MAML |
| | - | 300 | maximum # of input tokens |
| | - | 1 | # of demonstrations |
| | - | 1.0 | regulation weight in EWC |

# B Tokenizer Alignment: GPT and T5

When calling OpenAI's API to utilize text-davinci-003 (hereinafter referred to as GPT), we can get a predicted SQL query $\mathbf{p}_{0:L}$ made up of $L$ GPT tokens and a sequence of output distributions $\mathbf{s}_{1:L}$ where each $\mathbf{s}_i \in \mathbf{s}$ is the probabilities of the most probable tokens at each decoding step, corresponding to $\mathbf{p}_i$. Our objective is to convert the GPT output distributions to the the format of T5 tokenizer and get the target distributions $\mathbf{t}$ where each $\mathbf{t}_i \in \mathbf{t}$ contains the probabilities of T5 tokens, making it trainable for the student models. To achieve this goal, we apply the algorithm shown in Algorithm 1.

First, we evaluate whether GPT produces the correct SQL by comparing $\mathbf{p}$ with the gold SQL $\mathbf{g}$. If not, we expand $\mathbf{g}$ as a sequence of one-hot distributions for student's training. Otherwise, we tokenize the GPT predicted SQL query using the T5 tokenizer and get a sequence of one-hot distributions $\mathbf{q}_{1:K}$ made up of $K$ T5 tokens with probability 1. And our target is to align $\mathbf{s}$ and $\mathbf{q}$. Here we apply an one-to-one strategy. If there exists an one-to-one mapping of a T5 token $\mathbf{q}_i$ and a GPT token $\mathbf{p}_j$, we use the GPT distribution $\mathbf{s}_j$ as the T5 distribution. In other cases where a mismatch exists, we simply take the one-hot distribution $\mathbf{q}_i$ as the label.

---

**Algorithm 1** Tokenizer alignment of GPT and T5

---

**Require:** GPT predicted SQL query $\mathbf{p}_{1:L}$, GPT output distributions $\mathbf{s}_{1:L}$, gold SQL query $\mathbf{g}$,
      T5 Tokenizer $\mathbf{T}$.
  initialize $\mathbf{t} = [\,]$
  **function** FINDONETOONEMAPPING($\mathbf{q}_i$, $\mathbf{p}$, $\mathbf{s}$)
      **if** Exists an one-to-one mapping of $\mathbf{q}_i$ and $\mathbf{p}_j$ **then**
          **return** $\mathbf{s}_j$
      **else**
          **return** None
      **end if**
  **end function**
  **if** $\mathbf{p}$ is equivalent to $\mathbf{g}$ **then**
      $\mathbf{q}_{1:K} = \mathbf{T}$.TOKENIZE($\mathbf{p}$)
      **for** $i := 1$ **to** $K$ **do**
          **if** FINDONETOONEMAPPING($\mathbf{q}_i$, $\mathbf{p}$, $\mathbf{s}$) is not None **then**
              $\mathbf{t}$.APPEND(FINDONETOONEMAPPING($\mathbf{q}_i$, $\mathbf{p}$, $\mathbf{s}$))
          **else**
              $\mathbf{t}$.APPEND($\mathbf{q}_i$)
          **end if**
      **end for**
  **else**
      $\mathbf{t} = \mathbf{g}$
  **end if**
  return $\mathbf{t}$

---

