# OpenReview forum: "Parameterizing Context: Unleashing the Power of Parameter-Efficient Fine-Tuning and In-Context Tuning for Continual Table Semantic Parsing"
_NeurIPS.cc/2023/Conference — NeurIPS 2023 poster_

### Official Review · Reviewer_zd3d · 2023-06-26

**Soundness:** 2 fair
**Presentation:** 3 good
**Contribution:** 2 fair
**Rating:** 6
**Confidence:** 4

**Summary:**

The author presents a novel approach that combines PEFT and ICT to train a continual table semantic parser. The proposed solution is based on a teacher-student framework. Two task streams are developed from WikiSQL and Spider semantic parsing benchmarks. Experiments using C3 and baselines are conducted using these task streams. The results indicate that their method outperforms existing competitors, achieving state-of-the-art performance across multiple metrics.


**Strengths:**

- The fuse of ICT and PEFT continual framework is novel and effective on the proposed continual table semantic parsing task.
- The description of the framework and method is clearly written.


**Weaknesses:**

- The authors did not clearly state whether the Problem Formulation in Section 2.2 has been studied previously.
- The Continual table semantic parsing problem is not well motivated. For example, the authors claimed `After training on a new task, the performance of the parser on the previous task may plummet attributed to parameter updates.`, but there is no empirical evidence of this claim as a motivating example.
- The authors stated the "invisibility of past demonstrations" for ICT, which seems unrealistic even under data privacy concerns. ICT typically only requires a few examples, which can be obtained from public databases or synthetic/obfuscated database content.
- The authors can have additional baseline: Train on a reasonable size of public text2sql data with a mixture of domains and then zero-shot on the Stream datasets. The public data will not have privacy concerns. This will show the accuracy gap and rationale behind the continual training. It seems line 196 is similar to this, what is the training set size?


**Questions:**

- Is the Problem Formulation in Section 2.2 novel? Have other papers examined and studied this problem? If not, the problem formulation and its motivation should be part of the paper's contribution.
- In Table 1, what is the PEFT results on T5-large?
- In Figure 1, what are the blue arrows going into PLM in the ICT portion?
- In Figure 1, why are past demonstrations invisible? This seems to be an unrealistic setting, even under data privacy concerns. We can certainly have a diverse pool of demonstrations from public databases or synthetic/obfuscated database content.
- In Figure 3, what is the x-axis label?


**Limitations:**

No negative societal impacts

---

> ### Author Rebuttal · Authors · 2023-08-09
>
> We are very grateful to you for providing us with valuable feedback and suggestions for our paper. We will provide explanations and clarifications for each weakness and question.
>
> #### For Weakness 1:
> Before us, the problem formulation in $\rm{Section} \ 2.2$ was proposed by [7], so we did not include this as one of our contributions. However, their scenario is also slightly different from ours. They guarantee a portion of unsupervised data for each task on the stream, while this guarantee is removed in our scenario, making it more challenging.
>
> #### For Weakness 2:
> For this "the parser's performance on the previous task may drop drastically due to parameter updates" claim, we have actually shown it experimentally. Specifically, in $\rm{Figure} \ 4$, the Fine-Tuned parser exposes a significant performance degradation of about 20% in terms of TA (Task Accuracy) and EA (Example Accuracy) on both datasets as the task increases. The dramatic drop in MD (Memory Decay) metrics also proves that the model really has catastrophic forgetting about previous tasks.
>
> #### For Weakness 3:
>
> Public or synthetic/obfuscated databases are indeed a possible way to get demos. However, obtaining a valid demonstration may not be easy for two reasons:
> 1. First, a demonstration is an NLQ-SQL pair, and database content alone still makes it expensive to construct a large scale of natural language that matches human expressions, especially for some small domains.
> 2. Second, recent studies [23, 24] have shown that only in-context learning using demonstrations similar to the test samples can maximize performance gains. Unfortunately, for domain-specific databases, there is no guarantee that similar demonstrations will be found using public data resources. To verify this, we added an experiment: We assume that $D^0_\rm{train}$, the training set of task_0, is an always-visible public demonstration pool from which all demos for subsequent tasks are sampled. The following table shows the average accuracy of the Teacher Parser when it encounters each task at the first time.
>
> | **Method**                         | Spider-Stream (%) | Combined-Stream (%) |
> | ---------------------------------- | :---------------: | :-----------------: |
> | Teacher Parser (Demos from task_0) |       76.3        |        70.6         |
> | C3 Teacher Parser (Ours)           |       78.2        |        72.4         |
>
> From the results, when sampling the demonstration from $D^0_\rm{train}$, the performance of the Teacher parser shows a significant drop on both datasets. This again proves that the quality of the demonstration has a great impact on the performance of ICT.
>
>
> #### For Weakness 4:
> Yes, our setup at line 196 is trying to simulate this scenario. We randomly choose k (6 for Spider-stream and 5 for Combine-stream) tasks to merge as an initial task (task_0) for a mixed domain. For the Spider-stream, the size of the training set for task_0 is 2082; For the Combined-stream, the size of the training set for task_0 is 1373.
>
>
> #### For Question 1:
> Due to space limitations, please see response to Weakness #1 for details.
>
> #### For Question 2:
> We added a experiment to evaluate the performance of PEFT when using T5-Large. The experimental results are shown in the following table.
>
>
> | **Method**           | TA (%) | EA (%) | MD (%) | TA (%) | EA (%) | MD (%) |
> | -------------------- | :----: | :----: | :----: | :----: | :----: | :----: |
> | T5-Large + PEFT      |  69.7  |  67.4  |   0    |  67.3  |  70.0  |   0    |
> | T5-Large + C3 (Ours) |  70.7  |  68.9  |   0    |  69.0  |  71.2  |   0    |
>
>
> When using T5-Large as a backbone, our C3 still performs better than PEFT.
> If this paper is accepted, we'll add these results to the camera-ready version.
>
> #### For Question 3:
> Here the blue arrow indicates that the model should maintain performance on the inputs from the previous task even when it encounters subsequent tasks. To avoid misunderstandings, we will add an explanation in caption in subsequent releases.
>
> #### For Question 4:
> Due to space limitations, please see response to Weakness #3 for details.
>
> #### For Question 5:
> The x-axis represents the ID of the segmented task. Note that this is just the initial slice of the task (independent of the order), and as we mentioned before, when the methods are actually run, a random $k$ of them will constitute the initial task. The remaining $n - k$ tasks are shuffled as the task 2 to $n - k + 1$.

---

> > ### Comment · Reviewer_zd3d · 2023-08-16
> >
> > Thank you for your response. I have updated my rating

---

### Official Review · Reviewer_yC62 · 2023-07-03

**Soundness:** 3 good
**Presentation:** 3 good
**Contribution:** 2 fair
**Rating:** 5
**Confidence:** 3

**Summary:**

The paper introduces a method combining parameter-efficient fine-tuning (PEFT) and in-context tuning (ICT) to address the issues of overfitting and catastrophic forgetting in training a continual table semantic parser with limited training examples. Through a task-adaptive PEFT framework and a teacher-student setup that utilizes ICT, the method demonstrates enhanced performance in comparison to established baselines, as validated by experiments on two benchmarks.

**Strengths:**

1. This paper is overall good written and easy to follow.
2. This paper proposes a method that fuses PEFT with ICT to resolve the overfitting and catastrophic forgetting problem.
3. In addition to PEFT + ICT, the authors also propose a teacher-student framework that distills ICT results on teacher to a student model.

**Weaknesses:**

My main concern about this paper is its technical novelty.
1. Combining PEFT and ICT (demonstration) is not new. e.g., [Gao et al.](https://arxiv.org/abs/2012.15723)
2. Distilling soft prompts, on the other hand, is not new either. https://arxiv.org/abs/2212.10670 https://arxiv.org/abs/2304.08467
3. I found it unconvincing to exclude results from [7]. C3 also uses demonstration retrieval, which indicates the same pool of supervised data is used.
4. Following 3, the claim that the method is few-shot is inaccurate. Since the examplars are selected from a pool examples, comparing to methods that only use a few examples is unfair.

**Questions:**

### Questions
1. Why the authors choose tabular semantic parsing? Is this method generalizable to other tasks (e.g., general NLU/NLG tasks)?
2. Why use different backbones for other methods in Table 1? How do those methods perform on T5?

### Typos
Ln. 23: "finance [3], and ?"

---

> ### Author Rebuttal · Authors · 2023-08-09
>
> We are very grateful to you for providing us with valuable feedback and suggestions for our paper. We will provide explanations and clarifications for each weakness and question.
>
> #### For Weakness 1:
> Our method is fundamentally different from that of [Gao et al.] for the following two reasons:
> 1. [Gao et al.] is mainly based on hard prompt + ICT, which still requires fine-tuning the whole model rather than a subset of parameters during the training process, so it cannot be considered as a PEFT approach. In contrast, our method only updates soft prompt while freezing the whole PLM during the training of the teacher model, which reflects the parameter efficiency.
> 2. [Gao et al.] focuses on how to improve the few-shot learning capability of smaller pre-trained models on static datasets, whereas we focus on a dynamic task-stream scenario, exploring how to avoid catastrophic forgetting while guaranteeing few-shot learning capability.
>
> * Gao, Tianyu, Adam Fisch, and Danqi Chen. "Making pre-trained language models better few-shot learners." arXiv preprint arXiv:2012.15723 (2020).
>
> #### For Weakness 2:
> 1. Although [Huang et al.] also proposed in-conext learning distillation, their student model is essentially different from ours in terms of motivation. Their student model focuses only on the few-shot learning capability and still employs in-context learning for prediction without soft prompts. In contrast, our student model needs to handle dynamic task-stream scenarios. The invisibility of the past task context forces C3 to use PEFT to inject in-context learning capabilities into the soft prompts, avoiding catastrophic forgetting while improving few-shot learning.
>
> 2. The goal of [Mu et al.] is also not to solve the few-shot or continual learning problem, but to compress the prompt to improve LLM inference efficiency and save storage space. This is completely different from the dynamic task steam scenarios we've focused on.
>
> *  Huang, Yukun, et al. "In-context Learning Distillation: Transferring Few-shot Learning Ability of Pre-trained Language Models." arXiv preprint arXiv:2212.10670 (2022).
> * Mu, Jesse, Xiang Lisa Li, and Noah Goodman. "Learning to compress prompts with gist tokens." arXiv preprint arXiv:2304.08467 (2023).
>
> #### For Weakness 3:
> We did not compare with [7] because [7] defines a slightly different scenario than the one we are concerned with. They first assume that each task in the stream has corresponding unlabeled training set $ D^i_\rm{unsup}$, which contains only NLQs without gold SQL queries. Then, their method SFNet performs semi-supervised learning of the labeled training set $D^i_\rm{train}$ in combination with $D^i_\rm{unsup}$, whereas our method uses only $D^i_\rm{train}$. To ensure that all methods use the same training data, we did not compare with [7].
>
> However, to make the comparison more convincing, we added an experiment running [7] on Combined-Stream. The results are shown in the table below.
>
> | **Method**          | TA (%) | EA (%) | MD (%) |
> | ------------------- | :----: | :----: | :----: |
> | SFNet [7]           |  61.9  |  59.6  |  -3.2  |
> | T5-Base + C3 (Ours) |  67.7  |  66.7  |   0    |
>
> The reason why we only use Combined-stream and not on Spider-stream here is that Combined-stream only randomly selects a portion of samples from the original Spider and WikiSQL datasets, and the rest can be used as unsupervised data to fulfill the scenario requirements of [7].
>
> Despite utilizing additional unsupervised data for semi-supervised learning, [7] still does not perform as well as our proposed C3. Our method achieves better results under weaker assumptions.
>
> #### For Weakness 4:
> You may have some misunderstanding of our method. For $i$-th task, all the demonstrations of our C3 are sampled from the training set $D^i_\rm{train}$ (line 153), so the training set of C3 is still $D^i_\rm{train}$, which is consistent with all compared methods. So comparisons are fair.
> Our few-shot learning scenario setup follows existing work [8,9]. We adopt the standard $N$-way $K$-shot definition: "way" denotes the database and "shot" denotes the training example corresponding to the database.
> * For Spider-stream/Combined-Stream, there were an average of 337.8/292.7 training examples per task and 10.1/52.5 databases, i.e., 10-way 34-shot / 53-way 6-shot.
>
> We will include this setting to the future release.
>
>
> #### For Question 1:
> Yes. Theoretically, our proposed method is generalizable and has the potential to be adapted to other NLP tasks because it is not limited to specific backbone models and does not involve task-specific modules.
>
> We chose TSP for several reasons:
> 1. Databases are one of the most widely used information carriers, playing a fundamental role in various domains. As a key technology for natural language interfaces to databases, the study of TSP is of great significance for the enhancement of many fields. In addition, the natural continuous updating of databases leads to the task stream.
> 2. TSP is a difficult task. Unlike general NLU tasks (NER or RE), it requires the model to understand all aspects of the natural language query, including the entities (column names, table names) and the overall logical framework, and complete the mapping with the corresponding structured tables. Accordingly, the supervised data for this task is more difficult to obtain, resulting in the few-shot challenge.
> 3. Unlike NLG tasks (like summarization, machine translation), TSP is more objective in evaluating the result metrics. Any small error in the SQL will lead to completely incorrect results. We believe that this task can more rigorously reflect the true performance of our method.
>
> In future work, we will choose more different types of NLP tasks to evaluate our method.
>
> #### For Question 2:
> Due to space limitations, please see response to Weakness #1 mentioned by reviewer vuh7 for details.

---

### Official Review · Reviewer_vuh7 · 2023-07-06

**Soundness:** 3 good
**Presentation:** 3 good
**Contribution:** 2 fair
**Rating:** 5
**Confidence:** 4

**Summary:**

This paper introduces a novel method for training a continual table semantic parser, which aims to translate natural language into SQL based on task-specific tables with limited training examples. The proposed method integrates parameter-efficient fine-tuning (PEFT) and in-context tuning (ICT) to overcome the challenges of overfitting and catastrophic forgetting. The paper presents a task-adaptive PEFT framework and a teacher-student framework-based solution. Experimental evaluations demonstrate the superiority of the proposed method over existing baselines.

**Strengths:**

- The paper addresses an important problem in the field of table semantic parsing, namely the challenge of training a parser on a sequence of tasks with limited supervision.
- The proposed method integrates two existing techniques, PEFT and ICT, to overcome overfitting and catastrophic forgetting, respectively. This combination of methodologies is neat and straightforward.
- This paper provides a well-structured review of relevant literature, discussing prior work on table semantic parsing and related topics.
- The analysis is comprehensive and gives more insight on the future research.


**Weaknesses:**

- The baseline comparison appears somewhat unfair. While I appreciate the inclusion of multiple baselines by this paper, I noticed that most of them were implemented on GRAPPA to highlight their inferior performance, whereas this paper's proposed method was applied to T5 (base and large). Additionally, there is a lack of reported MD performance for both PEFT and C3, which should be the key metric to reflect the forgetting degree. This inconsistency in reporting the experimental setup leaves me somewhat confused about the evaluation process and the methodology employed in this paper.
- The results in Table 1 show that PEFT is dominant for catastrophic forgetting, while C3 is less dominant. in-context tuning does not play a prominent role here. And PEFT is already well-known for its advantage to avoid catastrophic forgetting, with the limitation that it cannot learn very different tasks fast.

**Questions:**

- I notice "Demo 2" appears twice in Figure 1. Is it a typo?
- I noticed that the paper utilizes the first dataset to initialize the model parameters for the entire model. This approach raises a natural concern: does it imply that the proposed method is only applicable to scenarios where the logical form is similar, but not when the logical form differs completely?

---

> ### Author Rebuttal · Authors · 2023-08-09
>
> We are very grateful to you for providing us with valuable feedback and suggestions for our paper. We will provide explanations and clarifications for each weakness and question.
>
> #### For Weakness 1:
> We chose GRAPPA as the backbone PLM for baselines for two main reasons:
> 1. Unlike T5, GRAPPA is pre-trained specifically for the TSP task (including Spider and WikiSQL), so we intuitively thought it would have stronger performance in our scenario.
> 2. Since some of the baselines we used predominantly applied PLMs with encoder-only architectures in the original papers, e.g., [8] (RoBERTa), [11] (BERT), we also followed them to use an encoder-only PLM (GRAPPA) in order to maintain the consistency with the original papers.
>
> In fact, by comparing the fine-tuning results in $\rm{Table} \ 1$ and $\rm{Table} \ 2$, it can be seen that GRAPPA performs better on Spider-stream. On the contrary, on Combined-stream, T5 has considerably less forgetfulness compared to Grappa's. We hypothesize that this is because T5's text-to-text transfer learning capability makes it less sensitive to significant differences in the logical form across tasks of Spider and WikiSQL.
>
> To ensure a fairer comparison, we added the experiments that apply the best-performing EMR and EMAR (in $\rm{Table} \ 1$ of our paper) to T5 as the new baselines. The experimental results are shown as follows:
>
>
> | Method              |            | Spider-Stream |            |            | Combined-Stream |            |
> | ------------------- | :--------: | :-----------: | :--------: | :--------: | :-------------: | :--------: |
> |                     | **TA (%)** |  **EA (%)**   | **MD (%)** | **TA (%)** |   **EA (%)**    | **MD (%)** |
> | T5-Base + EMR       |    59.1    |     56.7      |   -13.4    |    62.5    |      62.4       |    -5.7    |
> | T5-Base + EMAR      |    57.8    |     53.6      |   -15.7    |    62.0    |      59.7       |    -6.5    |
> | T5-Base + C3 (Ours) |    67.7    |     66.7      |     0      |    66.4    |      67.7       |     0      |
>
>
> From the results, C3 still has a significant improvement compared to the strong T5 baselines on both datasets. If this paper is accepted, we'll add these results to the camera-ready version.
>
> Note that the reason we did not add the MD performance of PEFT and C3 in $\rm{Table} \ 1$ and $\rm{Table} \ 2$ is because they have no forgetting, i.e., MD=0. Freezing PLM parameters ensures that they are not updated. Simply loading the checkpoint (soft prompt) for each task allows for an exact replication of the performance on previous tasks. We initially thought we should just omit it here because these cells will be constant values (zero). We regret that this omission misled you, and we will make a special note of it in the caption in subsequent version.
>
> #### For Weakness 2:
> In our scenario, catastrophic forgetting is more challenging compared to the few-shot problem. We add ICT's teacher model to PEFT precisely to enhance its lack of fast learning on new tasks. Although ICT's contribution to the overall performance is not as significant as PEFT, it is true that C3 achieves about 2% improvement over PEFT on both benchmarks in the case of multiple runs.
>
> #### For Question 1:
> Sorry, this is a typo, it should actually be `Demo 3`.
>
> #### For Question 2:
> In fact, we built Combined-Stream based on such a similar motivation as you mentioned. In Combined-Stream, the first task is based on Spider, while the second task is based on WikiSQL, and then alternates in turn. Spider has a complex SQL structure with syntax such as `JOIN`, `Nested Query`, `GROUP BY`, etc., whereas WikiSQL has only simple single-table SQL without complex syntax. To some extent, this setting can reflect the impact of the initial task on subsequent tasks. From the following three plots in $\rm{Figure} \ 4$, the performance oscillations in the baseline due to this stronger impact, which is effectively mitigated by our proposed method.

---

### Official Review · Reviewer_Bddu · 2023-07-07

**Soundness:** 3 good
**Presentation:** 3 good
**Contribution:** 3 good
**Rating:** 7
**Confidence:** 3

**Summary:**

The paper proposes a new continual learning method C3 for table semantic parsing which combines parameter-efficient fine-tuning (PEFT) and in-context tuning (ICT). The C3 framework contains a teacher network (ICT) that extracts contextual information from demonstrations and a student network (PEFT) that learns the teacher's output distribution. The ICT aims to enhance the ability of few-shot learning and PEFT aims to reduce catastrophic forgetting. The author construct two stream datasets from WikiSQL and Spider to evaluate their proposed method. C3 outperforms few-shot learning and continual learning baselines on these stream datasets.

**Strengths:**

1. The paper combines parameter-efficient fine-tuning (PEFT) and in-context tuning (ICT) with a teacher-student framework to leverage the unique advantages of each approach.  The proposed methods are effective in the continual table semantic parsing task.
2. The proposed framework is model-agnostic and seems generalizable to other continual learning tasks, further enhancing its value and applicability.

**Weaknesses:**

I don't see significant weakness but please address my concerns in Questions.

**Questions:**

1. In Table 3, when employing a T5-base model as the student model, it is observed that T5-large outperforms GPT-3 as the teacher model. However, interestingly, when the student model is upgraded to T5-large, the choice of teacher model seems less important. I am curious about the performance implications if the teacher model is completely removed when utilizing T5-large as the student model. I don't see this ablation study in Table 2.

---

> ### Author Rebuttal · Authors · 2023-08-09
>
> We are very grateful to you for providing us with valuable feedback and suggestions for our paper. We will provide explanations and clarifications for each question.
>
> #### For Question 1:
> We hypothesize that when the teacher model and student model share the same architecture (T5), the greater the difference in scale between the two, the more pronounced the distillation effect is.
>
> We explored the performance of C3 when removing the teacher model and using only T5-Large as the student model. The experimental results are shown in the following table.
>
> | Method               |            | Spider-Stream |            | |       Combined-Stream      |            |
> | -------------------- | :--------: | :-----------: | :--------: | :-------------: | :--------: | :--------: |
> |                      | **TA (%)** |  **EA (%)**   | **MD (%)** |   **TA (%)**    | **EA (%)** | **MD (%)** |
> | T5-Large + PEFT      |    69.7    |     67.4      |     0      |      67.3       |    70.0    |     0      |
> | T5-Large + C3 (Ours) |    70.7    |     68.9      |     0      |      69.0       |    71.2    |     0      |
>
> When using T5-Large as a backbone, removing the teacher model still results in a performance degradation compared to the entire C3.
> If this paper is accepted, we'll add these results to the camera-ready version.

---

> ### Comment · Reviewer_Bddu · 2023-08-18
>
> Thank you for your response!

---

### Comment · Area_Chair_rf67 · 2023-08-16
**check responses**

Dear reviewers, I appreciate your efforts in reviewing the submissions. However, we have not reached an agreement on the ratings yet. So please take a look at the responses from the authors and see if they have addressed your concerns/questions. Also, please read the reviews and responses from other reviewers to get a better understanding of the submissions. Thank you.

---

### Decision · Program_Chairs · 2023-09-21

**Decision:**

Accept (poster)

**Comment:**

This paper proposes a method that combines parameter-efficient fine-tuning (PEFT) and in-context tuning (ICT) to address the issues of catastrophic forgetting and overfitting for the table-based semantic parsing task. The teacher-student framework leverages ICT to extract contextual information from demonstrations and distills it to a student network that learns the teacher’s output distribution. The paper received mixed reviews from the 4 reviewers and the authors provided detailed answers, addressing most of the reviewers’ comments and questions. After reading the reviews, rebuttal, and discussions, I decided to recommend this paper an ACCEPT. In general, this paper presents an interesting and novel approach to table semantic parsing. Although the novelty of this paper (i.e., combining PEFT and ICT for table-based semantic parsing) was challenged by reviewer yC62, the authors have clearly explained the key differences between this work and previous work including Gao et al., 2020, Huang et al., 2022 and Mu et al., 2023. I highly recommend the authors to make these comparisons and explanations clearer in the revised version. Besides, as reviewer yC62 said, the proposed method can be seen as a general framework for most NLP tasks, instead of for table-based semantic paring only. So the authors should make it clear why table-based semantic paring is focused on here (as you did in the rebuttal).